# Molecular Structure-Based Prediction of Absorption Maxima of Dyes Using ANN Model

Neeraj Tomar [1] , Geeta Rani [2,*] , Vijaypal Singh Dhaka [2], Praveen K. Surolia [1,*] , Kalpit Gupta [2], Eugenio Vocaturo [3,4] and Ester Zumpano [3,4]

[1] Department of Chemistry, Manipal University Jaipur, Jaipur 303007, India; neerajtomar225@gmail.com
[2] Department of Computer and Communication Engineering, Manipal University Jaipur, Jaipur 303007, India; vijaypalsingh.dhaka@jaipur.manipal.edu (V.S.D.); kalpitgupta369@gmail.com (K.G.)
[3] Department of Computer Engineering, Modeling, Electronics and Systems, University of Calabria, 87036 Rende, Italy; e.vocaturo@dimes.unical.it or eugenio.vocaturo@cnr.it (E.V.); e.zumpano@dimes.unical.it (E.Z.)
[4] CNR NANOTEC, Via Pietro Bucci 33C, 87036 Arcavacata, Italy
* Correspondence: geetachhikara@gmail.com (G.R.); praveenkumar.surolia@jaipur.manipal.edu (P.K.S.)

**Abstract:** The exponentially growing energy requirements and, in turn, extensive depletion of non-restorable sources of energy are a major cause of concern. Restorable energy sources such as solar cells can be used as an alternative. However, their low efficiency is a barrier to their practical use. This provokes the research community to design efficient solar cells. Based on the study of efficacy, design feasibility, and cost of fabrication, DSSC shows supremacy over other photovoltaic solar cells. However, fabricating DSSC in a laboratory and then assessing their characteristics is a costly affair. The researchers applied techniques of computational chemistry such as Time-Dependent Density Functional Theory, and an ab initio method for defining the structure and electronic properties of dyes without synthesizing them. However, the inability of descriptors to provide an intuitive physical depiction of the effect of all parameters is a limitation of the proposed approaches. The proven potential of neural network models in data analysis, pattern recognition, and object detection motivated researchers to extend their applicability for predicting the absorption maxima ($\lambda_{max}$) of dye. The objective of this research is to develop an ANN-based QSPR model for correctly predicting the value of $\lambda_{max}$ for inorganic ruthenium complex dyes used in DSSC. Furthermore, it demonstrates the impact of different activation functions, optimizers, and loss functions on the prediction accuracy of $\lambda_{max}$. Moreover, this research showcases the impact of atomic weight, types of bonds between constituents of the dye molecule, and the molecular weight of the dye molecule on the value of $\lambda_{max}$. The experimental results proved that the value of $\lambda_{max}$ varies with changes in constituent atoms and types of bonds in a dye molecule. In addition, the model minimizes the difference in the experimental and calculated values of absorption maxima. The comparison with the existing models proved the dominance of the proposed model.

**Keywords:** solar; DSSC; artificial neural network; energy; $\lambda_{max}$

## 1. Introduction

Electricity consumption is increasing proportionally with an increase in population. Mankind mainly depends on non-restorable energy sources such as coal and fossil fuels for electricity production [1]. These non-restorable sources will be exhausted in the future if depletion continues at the same rate. Furthermore, these sources cause environmental pollution. Therefore, researchers emphasize designing the devices to harness the energy from renewable sources such as biomass, wind, hydroelectric, geothermal, and solar energy [2]. Electricity production utilizing solar energy is cleaner and safer than conventional sources. In the recent era, Photovoltaics (PV) technology is considered the most encouraging technology due to its potential to convert solar energy into electrical energy [3]. The PV cells

developed so far have been categorized into three generations. The cells designed in the first generation consist of monocrystalline and polycrystalline silicon. The PV cells of the second generation consist of silicon of non-crystalline form, cadmium telluride, and copper gallium indium diselenide. Along with the advantages of the first and second generation of solar cells in their better performance, there are certain limitations. The materials used in the development of the first and second generation of solar cells are hazardous and expensive. To conquer these issues, scientists have developed third-generation solar cells, such as Dye-Sensitized Solar Cells (DSSC), quantum Dot (QDs) organic, and Perovskite Solar Cells (PSC) [4–7]. Based on the study of efficacy, design feasibility, and cost of fabrication, DSSC shows supremacy over other PV cells developed in the first and second-generation [8–10]. Furthermore, DSSC is attractive to industry and users due to its high molar absorption coefficient, and potential to perform under diffused light conditions. Moreover, DSSC has low fabrication cost, is processable at ambient temperature, easy to manufacture, and suitable for roll-to-roll production. Further, the material's ecofriendly nature, printability on a flexible substrate, and availability in a variety of colors increase the importance of DSSC in real-life [11,12].

DSSC is an integration of components viz. photoanode, with a semiconductor layer, dye sensitizer, electrolyte and counter electrode with a thin layer of catalyst [13]. Along with all components, the dye plays an important role in deciding the efficiency of a DSSC because it is responsible for the absorption of photons from the incident sunlight [14,15]. It is covalently bonded to semiconductor oxide. These dyes have been extensively tested in the fabrication of DSSC and are classified into three groups based on their source or components used for manufacturing. For example, dyes extracted from plant parts such as fruits, flowers, and leaves are considered natural dyes [16]. Dyes fabricated by using metal complexes such as ruthenium [8,17], osmium [18], platinum [19], copper [20], iridium [21], etc. are classified as metal complex dyes. Metal-based dyes are preferred in DSSC due to their advanced photo-conversion efficiency. In contrast, the metal-free organic dyes were introduced at a later stage due to their low cost, high molar extinction coefficient, and simple fabrication technique [22,23]. However, metal-free dyes still show less photovoltaic efficiency compared to metal-complex dyes. Among natural, organic and inorganic dyes, the inorganic dyes' mainly polypyridyl complex of ruthenium metal has been widely used and investigated [24]. Inorganic dyes are selected for their high stability and excellent redox properties [25]. Further, an efficient sensitizer satisfies the following five conditions.

(i)    The bond between the semiconductor oxide surface and dye must be strong enough to move the electron injection in the Conduction Band (CB) of the semiconductor oxide.
(ii)    The LUMO of the sensitizer should be greater than $TiO_2$ CB. It empowers the charge injection.
(iii)    The molecule of dye must be small because the bulky molecule can lead to a lower optical cross-section.
(iv)    The dye must be thermally, photochemically, and electrochemically vigorous. If the oxidation-back reduction turnover number exceeds 106, then the stability of DSSC can reach up to approximately 20 years.
(v)    The sensitizer should be effective in absorbing all light below the 920 nm wavelength strike to the surface of the semiconductor oxide [2].

The above-discussed conditions are indicative of the challenging synthesis of such an efficient and novel dye sensitizer that includes all the above-mentioned characteristics. The hit and trial experiments in the laboratory incur a high cost, require expertise in the synthesis of DSSC, and consume a lot of time. Thus, fabricating DSSC in the laboratory and then assessing their characteristics is a costly affair. Therefore, there is a strong need to find an alternative that minimizes the cost and time for trial experiments.

The researchers apply computational chemistry in defining the structure and electronic properties of dyes without actually synthesizing them. For example, the Time-Dependent Density Functional Theory (TD-DFT) [26] and an ab initio method [27] have been employed for identifying new organic dyes for synthesizing DSSC. TD-DFT is preferred for investi-

gating the properties of organic dyes in their excited state due to its higher accuracy and lower computational time than the ab initio method [28,29].

To further improve the prediction accuracy, Xu et. al. employed the QSPR model using Polak–Ribiere algorithm in HYPERCHEM for the prediction of absorption maxima ($\lambda_{max}$) of organic dyes [30]. They employed DRAGON software to calculate three-dimensional (3-D) descriptors from the optimized molecular geometries. In the subsequent research work, Xu et.al. designed a QSPR between descriptors [31]. They represented that the molecular structures and the $\lambda_{max}$ of organic dyes used in DSSC follow the same protocols as applied by Colombo et al. in [20]. The disadvantage of the QSPR approach is that the descriptors do not always provide an intuitive physical depiction of the effect of all parameters [32].

Further, to develop a nonlinear model, researchers applied a quasi-Newton Broyden–Fletcher–Goldfarb–Shanno (BFGS) algorithm [31,33]. They applied the algorithm on the same dataset as used in the research works discussed in [30,31,34]. The details of the dataset are shown in Table S1.

In the BFGS algorithm, there is no need to specify the rate or momentum. Furthermore, it undergoes fast training. However, it is unable to determine small and medium scale minimizing functions. It requires a large amount of memory, and therefore, it involves a huge extent of numerical operations [35].

These challenges can be resolved by employing the Artificial Neural Networks (ANN) models [36,37]. Although the potential of neural network models in data analysis [38], pattern recognition [39], and object detection [40] is proven in various application areas such as healthcare [41–43], agriculture [44], and material science [45], only a few researchers employed the ANN-based models for predicting the absorption maxima ($\lambda_{max}$) of dye [30,31]. Thus, there is a huge scope to extend their applicability. In this research, we propose an ANN-based QSPR model for correctly predicting the value of $\lambda_{max}$ for inorganic ruthenium complex dyes used in DSSC.

The major objectives of this research are as follows.

(i)    To develop an ANN-based model for predicting the absorption maxima of the dye sensitizer used in DSSC.
(ii)   To minimize the difference in the experimental and calculated values of absorption maxima.
(iii)  To showcase the impacts of the atomic weight of each atom and molecular weight on the value of $\lambda_{max}$.
(iv)   To demonstrate the impact of different types of bonds on the value of $\lambda_{max}$.
(v)    To justify the impact of different activation functions, optimizers, and loss functions on the prediction accuracy of $\lambda_{max}$ using the ANN model.

The structure of the article is as follows: Section 1 provides the introduction. It gives an overview of the research topic, introduces the research problem, highlights the gaps in existing knowledge, and presents the objectives of the study. Section 2 describes the data collection and methodology of the research work. Section 3 illustrates the results. It presents the findings of the study. It includes tables, figures, and statistical analyses to support the findings. Section 4 presents the discussion of the research work. It interprets and analyzes the results, relates them to the objectives, and compares them with previous research. Section 5 presents the conclusion of the work. It summarizes the main findings and their implications. It also offers insights for future investigations.

## 2. Materials and Methods

### 2.1. Data Set

To prepare the dataset, the molecular structures of 81 ruthenium dye complexes were taken from the literature. From these structures of ruthenium dye complexes, molecular weight, atomic weight, number of all types of bonds such as C-C, C=C, C-N, Ru-N, C=O, Ru-NCS, C-O, and other bonds were calculated for each dye. The prepared dataset comprises 81 rows and 15 columns. The sample dataset is shown in Table 1 and the complete dataset is shown in Table S1. As reported in the earlier works [46,47], the experimental

values of $\lambda_{max}$ are also dependent on the solvent. Therefore, to ignore the impact of solvent, the dataset is collected for a single solvent viz. Dimethylformamide (DMF). It is obvious from the sample dataset shown in Table 1 that the values of $\lambda_{max}$ lie in the range from 473 to 631 nm in the collected dataset. This large variation in the range of values is important for improving the robustness of the ANN model. It means that the performance of the model does not degrade with any change in the value of $\lambda_{max}$. So, the model works efficiently for a wide range of dyes to correctly predict the value of $\lambda_{max}$.

Further, the number of each type of bond was inferred from the structure of dyes. These are shown as $N^+Bu_4$ =2, O-H=1, O-Na=1, O-H=2, C-S=4, C-S=8, C-Se=4, C-S=12, $N^+(C4H9)_4$=1, C-S=2, O-H=3, C-F=3, N≡N=1, C-F=6, O-H=7, N≡N=2, $TBA^+$=1, N-H=2, N-H=4, O-H=4, and $TBA^+$=1. Here, the symbol shows the type of bond, and the digit denotes the number of bonds or functional groups present in a dye molecule. For example, $N^+Bu_4$ =2 means that the dye contains two $N^+Bu_4$ groups, and O-H=1 means that the dye contains one O-H bond. The other groups can be interpreted in the same way.

*2.2. Experiments*

2.2.1. Architecture of Model

An Artificial Neural Network (ANN) is a machine learning model that is inspired by the structure and function of biological neurons in the brain. An ANN consists of multiple interconnected nodes i.e., neurons, organized into layers. Each neuron in the network has a set of weights associated with it, which determine the strength of its connections to other neurons in the network. The input layer of an ANN receives input data, which is then passed through one or more hidden layers. Each hidden layer applies a non-linear transformation to the input. During the training phase, its neurons adjust the weights to minimize the difference (value of loss function) between the predicted output and the actual output. Finally, the output layer of the network provides the prediction.

Rather than using the ANN models available in the literature [30,31], a customized ANN-based shallow network has been designed in this research. Its architecture is shown in Figure 1.

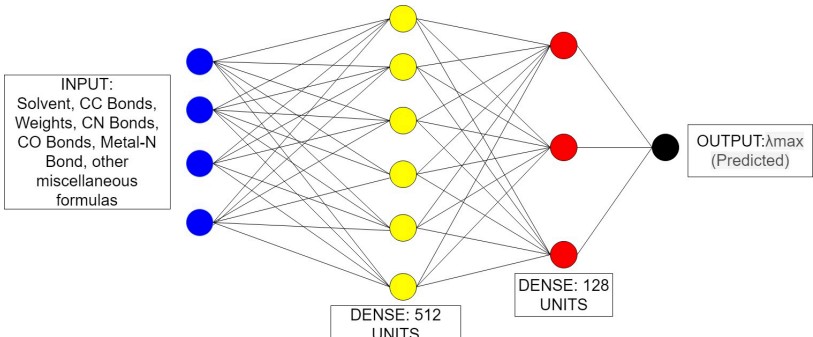

**Figure 1.** Sequential ANN Model.

The designed model can deal with observable outliers available in the data. Furthermore, the model requires a smaller dataset for training. Before, the final selection of the architecture, the ANN-based architectures with dense layers comprising 16, 32, 64, and 1024 Units were implemented on a trial-and-error basis. Furthermore, the experiments by employing different activation functions viz. Leaky ReLu, ReLu, Softmax, and Sigmoid; optimizers viz. Adam, AdaGrad, SGD, and RMSProp; and loss functions viz. Mean Absolute Error, Mean Square Error, Mean Squared Logarithmic Error, Binary Cross-Entropy, and Kullback Leibler Divergence Error were employed for experiments. The experimental results obtained by employing the above-mentioned parameters are shown in the subsequent Section 2.2. The impacts of these parameters on the prediction accuracy justify the selection of the 'Relu' activation function, Mean Absolute Error (MSE) loss function, and 'Adam' optimizer in the proposed ANN architecture.

**Table 1.** Collected and calculated data of Ruthenium dyes.

| Dye | Structure | Formula | Mol. Weight | Atomic Weight | $\lambda_{max}$ (MLCT) | Solvent | C-C Bond | C=C Bond | C-N Bond | Metal-N Bond | C=O Bond | Metal-NCS Bond | C-O Bond | Other Bonds/ Groups | Ref. |
|---|---|---|---|---|---|---|---|---|---|---|---|---|---|---|---|
| N749 |  | C69H116N9O6RuS3 | 1364.98 | C-828.74 H-116.92 N-126.06 O-96 Ru-101.07 S-96.19 | 600 | DMF | 11 | 6 | 6 | 3 | 3 | 3 | 3 | $N^+Bu_4$=2 | [48] |
| N719 |  | C58H86N8O8RuS2 | 1188.55 | C-696.62 H-86.68 N-112.05 O-128.00 Ru-101.07 S-64.13 | 525 | DMF | 13 | 8 | 8 | 4 | 4 | 2 | 4 | N+Bu4 =2 | [49,50] |
| Z907 |  | C42H52N6O4RuS2 | 870.10 | C-504.45 H-52.41 N-84.04 O-64 Ru-101.07 S-64.13 | 520 | DMF | 30 | 8 | 8 | 4 | 2 | 2 | 2 | O-H=1 O-Na=1 | [51] |
| YS-1 |  | C58H48N6O4RuS2 | 1058.24 | C-696.62 H-48.38 N-84.04 O-64 Ru-101.07 S-64.13 | 536 | DMF | 40 | 16 | 8 | 4 | 2 | 2 | 2 | O-H=2 | [51] |
| YS-2 |  | C74H80N6O4RuS2 | 1282.66 | C-888.79 H-80.63 N-84.04 O-64 Ru-101.07 S-64.13 | 536 | DMF | 56 | 16 | 8 | 4 | 2 | 2 | 2 | O-H=2 | [51] |

The proposed ANN model comprises one input layer, two dense layers, and one output layer. The first and second dense layers comprise 512 and 128 units, respectively. Further, the model contains 20,480; 65,664 parameters at the first and second dense layers, respectively. The number of trainable parameters was reduced to 129 at the output layer. This shows that the employed ANN model involves 86,273 trainable parameters. It does not involve any non-trainable parameter in its architecture. The model is trained with a batch size of 40 for 1000 epochs. Its efficacy is evaluated by using the evaluation metrics defined below.

i    Difference $\lambda_{max}$: This is the difference in the predicted and experimental value of absorption maxima, as defined in Equation (1):

$$\textbf{Difference } \boldsymbol{\lambda\textbf{max}} = \textbf{Predicted } \boldsymbol{\lambda\textbf{max}} - \textbf{Experimental } \boldsymbol{\lambda\textbf{max}} \tag{1}$$

ii    **Percentage error (Error%):** This is the percentage of difference in the predicted and experimental value of absorption maxima, as defined in Equation (2).

$$\text{Error \%} = \frac{\textbf{Difference } \boldsymbol{\lambda\textbf{max}}}{\textbf{Experimental } \boldsymbol{\lambda\textbf{max}}} \times \textbf{100} \tag{2}$$

iii    **Correlation matrix:** This matrix shows the correlation between (i) $\lambda_{max}$ and all bonds in dye molecule (ii) $\lambda_{max}$ and other additional groups present in a dye structure (iii) $\lambda_{max}$ and atomic and molecular weight. The matrix represents the direct as well as inverse correlation. The value '0' denotes no correlation, '1' indicates complete and direct correlation. Whereas '−1' shows that the given parameters have a complete and inverse correlation. The values increasing from 0 to 1 show an increasing degree of direct correlation. On the other hand, values approaching from 0 to −1 indicate the increasing degree of negative correlation between the parameters.

2.2.2. Selection of Hyperparameters

In this sub-section, the experiments conducted to select the optimum parameters are demonstrated.

Selection of Activation Function

Activation functions are employed in the neural networks to introduce non-linearity and enabling them to learn complex patterns in the input data. In this research, we employed the ReLU (Rectified Linear Unit) activation function. It is a simple and computationally efficient function that sets all negative values in the input to zero and leaves positive values unchanged as defined in Equation (3).

$$f(x) = max(0, x) \tag{3}$$

Here, $x$ is the input to the function, and $f(x)$ is the output. The ReLU function returns the input $x$, if it is positive, and returns 0 otherwise. This makes the ReLU function a simple yet powerful way to introduce non-linearity into neural networks.

The selection of ReLU activation function is based on the set of experiments conducted. The performance of ANN by employing different activation functions viz. Leaky ReLU, ReLU, Sigmoid, and Softmax are demonstrated in Figure 2. The difference in the predicted and experimental values of absorption maxima was observed. Further, the percentage error was calculated by employing the above-mentioned activation functions. It is evident from the results demonstrated in Figure 2 that employing the ReLu activation function in the proposed ANN model reports the minimum, whereas the softmax activation function results in the maximum percentage error in predicting of $\lambda_{max}$. Therefore, the ReLu activation function was employed in this research.

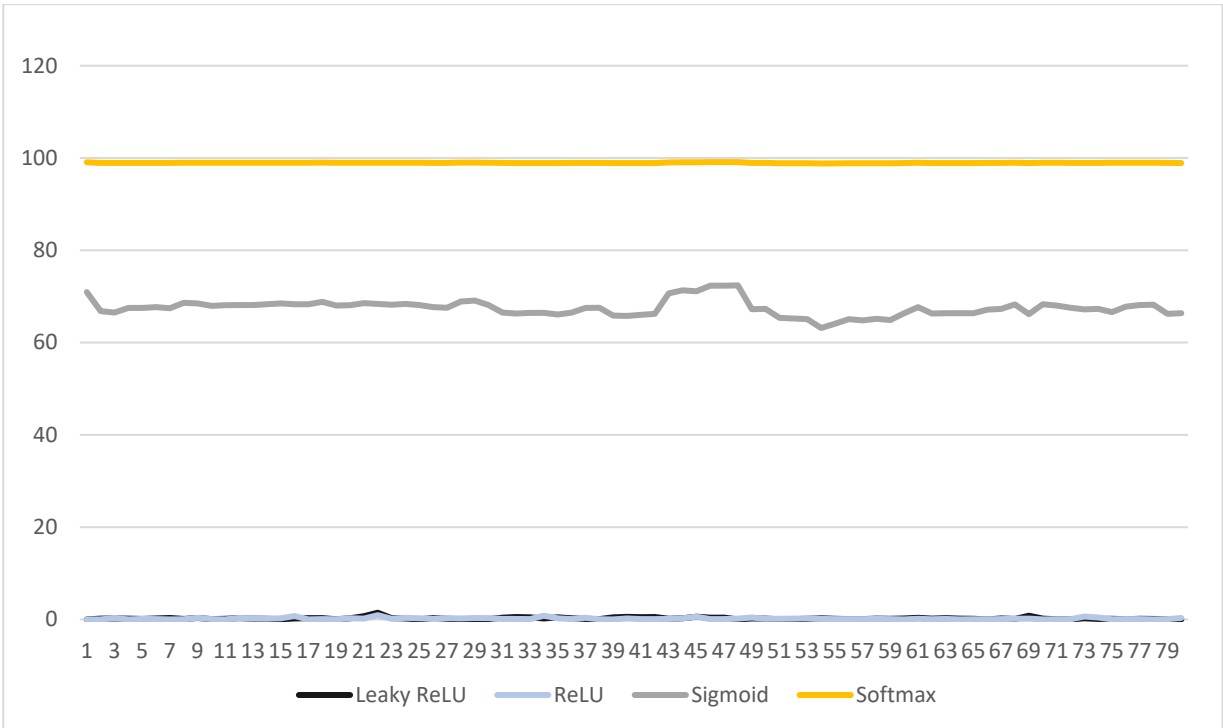

**Figure 2.** Comparison in percentage error reported by activation functions.

Selection of Loss Function

In this work, the selection of loss functions is accomplished strategically. Initially, the loss functions viz. mean absolute error, mean squared error, mean squared logarithmic error, categorical cross entropy and Kullback–Leibler divergence error were employed individually for predicting the value of $\lambda_{max}$. The values of percentage error in the $\lambda_{max}$ obtained for each loss function were evaluated. It is evident from the results demonstrated in Figure 3 that the mean absolute error reports the minimum value of percentage error. Thus, this loss function is employed in the architecture of the proposed ANN model.

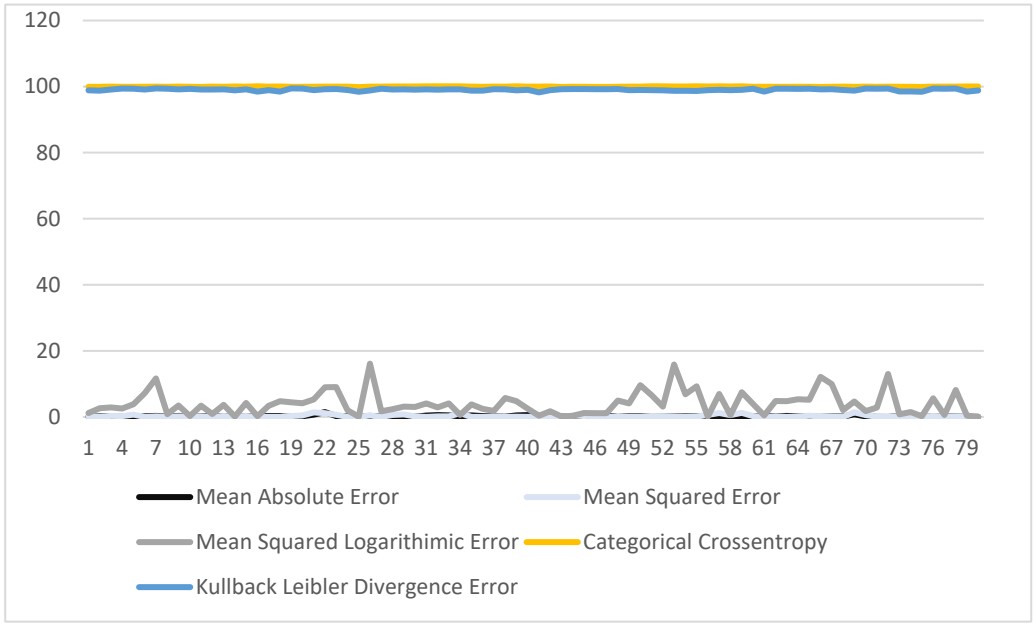

**Figure 3.** Percentage error comparison for loss functions.

Selection of Optimizer

Selecting the most suitable optimizer plays an important role in improving the prediction accuracy and minimizing the percentage error. For selecting the appropriate optimizer for the proposed model, a series of experiments were conducted. The optimizers, namely Adam, SGD, RMSProp, AdaGrad, were employed individually and the values of percentage error in the $\lambda_{max}$ were recorded. It is clear from the results shown in Figure 4 that the Adam optimizer results in the minimum value of percentage error. Therefore, the Adam optimizer was employed with the proposed ANN model.

**Figure 4.** Percentage error comparison for optimizer functions.

### 3. Results

The proposed ANN model is trained for 1000 epochs. The results of the trained model were recorded on the validation and testing datasets. The predicted values of absorption maxima based on the structure of the dye molecule, numbers of bonds, molecular weight, and atomic weight are demonstrated in the correlation matrices shown in Figures 5–9. The details of the correlation obtained are discussed below. Two more machine learning algorithms 'XGBoost', and random forest were applied, and their performance is compared with the ANN model. The comparative analysis is demonstrated in Figures 10 and 11.

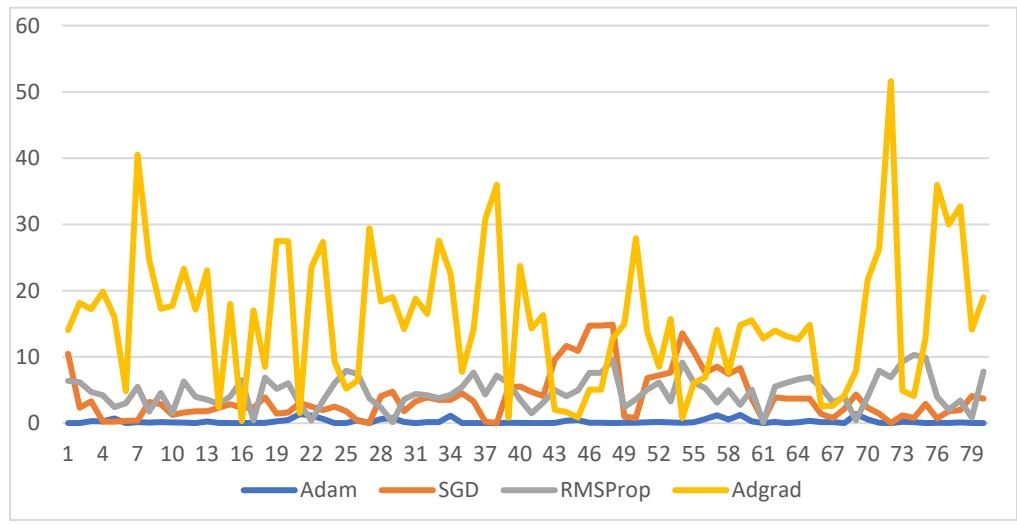

**Figure 5.** Correlation matrix for $\lambda_{max}$ and C-C, C=C, C-N, C=O, Metal-NCS, C-O, Metal-N bonds.

| λmax (MLCT) | 1 | 0.1304 | 0.1 | 0.152 | 0.192 | -0.0068 | -0.065 | 0.05 |
|---|---|---|---|---|---|---|---|---|
| N+Bu4 =2 | 0.13 | 1 | -0.07 | -0.06 | -0.19 | -0.0727 | -0.053 | -0.02 |
| O−H=1 | 0.10 | -0.073 | 1 | 0.867 | -0.54 | 0.13646 | 0.068 | 0.24 |
| O−Na=1 | 0.15 | -0.063 | 0.87 | 1 | -0.47 | 0.1 | 0.11 | 0.28 |
| O−H=2 | 0.19 | -0.19 | -0.54 | -0.47 | 1 | 0.058 | 0.11 | -0.13 |
| C−S=4 | -0.0068 | -0.073 | 0.14 | 0.105 | 0.058 | 1 | -0.15 | -0.05 |
| C−S=8 | -0.0651 | -0.053 | 0.07 | 0.11 | 0.114 | -0.15 | 1 | -0.04 |
| C−Se=4 | 0.045 | -0.018 | 0.24 | 0.28 | -0.13 | -0.051 | -0.037 | 1 |
| | λmax (MLCT) | N+Bu4 =2 | O−H=1 | O−Na=1 | O−H=2 | C−S=4 | C−S=8 | C−Se=4 |

**Figure 6.** Correlation matrix for $\lambda_{max}$ and N$^+$Bu$_4$=2, O-H=1, O-Na=1, O-H=2 C-S=4, C-S=8, and C-Se=4.

| λmax (MLCT) | 1 | -0.017 | 0.034 | 0.37 | -0.26 | -0.17 | -0.11 | -0.52 |
|---|---|---|---|---|---|---|---|---|
| C−S=12 | -0.017 | 1 | -0.0178 | -0.063 | -0.056 | -0.0527 | -0.03 | -0.073 |
| N+(C4H9)4=1 | 0.034 | -0.018 | 1 | -0.044 | -0.04 | -0.037 | -0.02 | -0.051 |
| C−S=2 | 0.37 | -0.063 | -0.0443 | 1 | 0.204 | 0.11035 | 0.17 | 0.0094 |
| O−H=3 | -0.26 | -0.056 | -0.0395 | 0.20 | 1 | 0.67 | 0.45 | 0.046 |
| C−F=3 | -0.17 | -0.053 | -0.037 | 0.11 | 0.67 | 1 | 0.48 | -0.15 |
| N−N=1 | -0.11 | -0.025 | -0.0178 | 0.17 | 0.45 | 0.48 | 1 | -0.073 |
| C−F=6 | -0.52 | -0.073 | -0.0511 | 0.0094 | 0.046 | -0.15 | -0.073 | 1 |
| | λmax (MLCT) | C−S=12 | N+(C4H9)4=1 | C−S=2 | O−H=3 | C−F=3 | N−N=1 | C−F=6 |

**Figure 7.** Correlation matrix for $\lambda_{max}$ and C-S, N$^+$(C$_4$H$_9$), O-H, C-F, N≡N bonds.

| λmax (MLCT) | 1 | -0.092 | -0.36 | 0.66 | 0.00444 | -0.031 | -0.059 | -0.081 |
|---|---|---|---|---|---|---|---|---|
| O−H=7 | -0.092 | 1 | -0.037 | -0.0344 | -0.0125 | -0.018 | -0.013 | -0.013 |
| N−N=2 | -0.36 | -0.037 | 1 | -0.1 | -0.037 | -0.053 | -0.037 | -0.037 |
| TBA+=1 | 0.66 | -0.034 | -0.1 | 1 | -0.0344 | -0.049 | -0.034 | -0.034 |
| N−H=2 | 0.004 | -0.012 | -0.037 | -0.0344 | 1 | -0.018 | -0.012 | -0.013 |
| N−H=4 | -0.031 | -0.018 | -0.0527 | -0.0489 | -0.0178 | 1 | -0.018 | -0.018 |
| O−H=4 | -0.059 | -0.013 | -0.037 | -0.0344 | -0.0125 | -0.018 | 1 | -0.013 |
| TBA=1 | -0.081 | -0.013 | -0.037 | -0.0344 | -0.0125 | -0.018 | -0.013 | 1 |
| | λmax (MLCT) | O−H=7 | N−N=2 | TBA+=1 | N−H=2 | N−H=4 | O−H=4 | TBA=1 |

**Figure 8.** Correlation matrix for $\lambda_{max}$ and O-H, N≡N, TBA$^+$, N-H bonds.

| | λmax (MLCT) | C_Weight | H_Weight | N_Weight | O_Weight | Ru_Weight | S_Weight | Na_Weight | Se_Weight | F_Weight | Molecular Weight |
|---|---|---|---|---|---|---|---|---|---|---|---|
| **λmax (MLCT)** | 1 | 0.16 | 0.36 | -0.32 | 0.098 | | 0.44 | 0.16 | 0.045 | -0.58 | 0.17 |
| **C_Weight** | 0.16 | 1 | 0.75 | 0.37 | 0.14 | | 0.26 | 0.03 | -0.07 | -0.34 | 0.95 |
| **H_Weight** | 0.36 | 0.75 | 1 | 0.21 | 0.317 | | 0.23 | -0.04 | -0.05 | -0.42 | 0.78 |
| **N_Weight** | -0.32 | 0.37 | 0.21 | 1 | 0.094 | | -0.36 | -0.01 | -0.1 | 0.39 | 0.42 |
| **O_Weight** | 0.098 | 0.14 | 0.32 | 0.09 | 1 | | -0.02 | 0.24 | -0.07 | -0.03 | 0.27 |
| **Ru_Weight** | | | | | | | | | | | |
| **S_Weight** | 0.44 | 0.26 | 0.23 | -0.36 | -0.02 | | 1 | 0.07 | -0.04 | -0.53 | 0.33 |
| **Na_Weight** | 0.16 | 0.03 | -0 | -0.01 | 0.242 | | 0.07 | 1 | -0.03 | -0.15 | 0.05 |
| **Se_Weight** | 0.045 | -0.07 | -0 | -0.1 | -0.07 | | -0.04 | -0.03 | 1 | -0.06 | -0 |
| **F_Weight** | -0.58 | -0.34 | -0.4 | 0.39 | -0.03 | | -0.53 | -0.15 | -0.06 | 1 | -0.25 |
| **Molecular Weight** | 0.17 | 0.95 | 0.78 | 0.42 | 0.266 | | 0.33 | 0.05 | -0.02 | -0.25 | 1 |

**Figure 9.** Correlation matrix for $\lambda_{max}$ based on atomic and molecular weight.

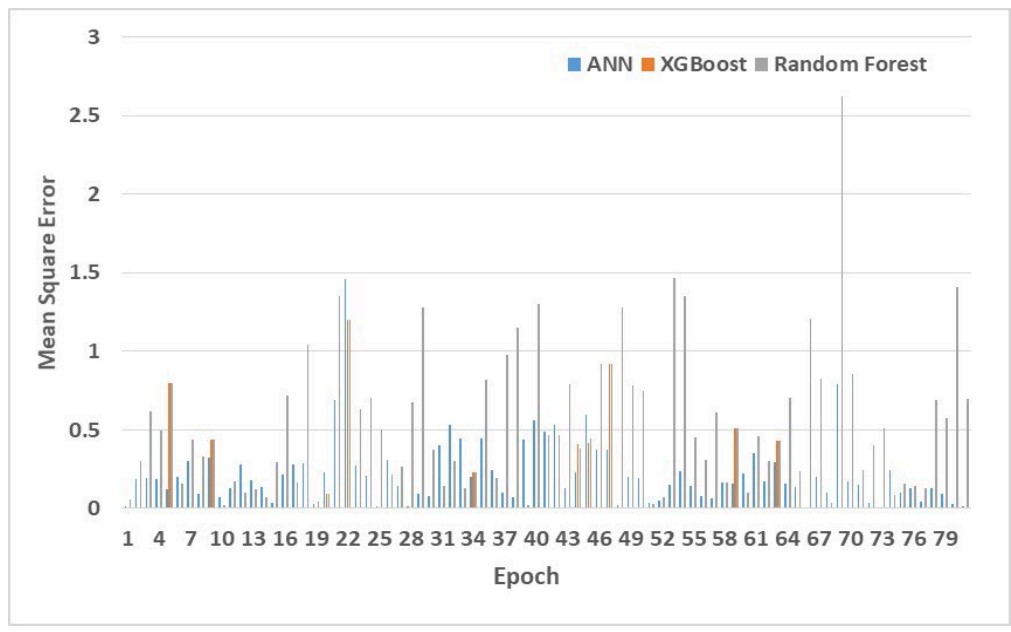

**Figure 10.** Comparison in MSE of ANN, XGBoost, and Random Forest Models.

### 3.1. Prediction of $\lambda_{max}$ Based on the C-C, C=C, C-N, C=O, Metal-NCS, C-O, Metal-N Bonds

The correlation between $\lambda_{max}$ and different types of bonds is demonstrated in Figure 5. The correlation studied from the correlation matrix is observed as linear. The range of $\lambda_{max}$ varies from 1 to −1. Values of 1 or close to 1 indicate a higher positive correlation, whereas the value '−1' or close to '−1' denote a negative correlation between the considered parameters. For example, it is evident from the sixth row and first column of the correlation chart shown in Figure 5 that the metal-NCS bond reports the value 0.77 which is close to 1. It shows the highest correlation between the metal-NCS bond and the value of $\lambda_{max}$. Furthermore, it is evident from its positive value that the increase in the metal-NCS bond leads to an increase in the value of $\lambda_{max}$ proportionally.

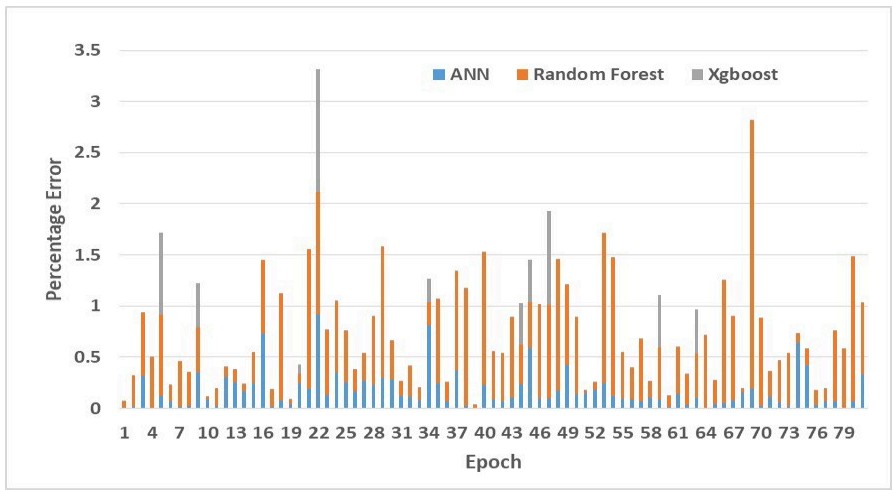

**Figure 11.** Comparison in percentage error reported by ANN, Xgboost, and Random Forest.

Further, the value '−0.74' shown in the eighth row, the first column of the correlation matrix indicates the high inverse correlation of metal-N bond with the value of $\lambda_{max}$. This means that the increase in the number of metal-N bonds in a dye leads to a decrease in the value of $\lambda_{max}$. Similarly, the value '−0.48' in the fourth row and first column shows that $\lambda_{max}$ and C–N bond are inversely correlated but the degree of correlation is lower than a metal-N bond. Next, the observation from column one of the second and ninth row shows that the C–C bond and DMF have a negligible correlation with the value of $\lambda_{max}$.

Additionally, the correlation matrix also shows the correlation between different bonds and functional groups present in a dye. For example, the value '0.54' recorded in the third column, the second row indicates that the number of C=C increases with an increase in the number of C-C single in the molecular structure of a dye. In contrast, the correlation of C-C with C-N is 0.38, C=O is 0.25, the metal-NCS bond is 0.18 and C-O is 0.11. These values are too small to have an impact on each other.

Similarly, the inter-correlation of C-N with C-C bond is 0.38, C=C bond is 0.47, C=O is 0.26, metal-NCS is −0.47, C-O is −0.19, and with the metal-N bond is 0.49. It reflects that the C-N bond has the highest correlation with the metal-N bond among all the above-stated bonds.

Next, the C=O bond is correlated to C-C, C=C, C-N, metal-NCS, C−O, and metal-N bonds with the values 0.25, −0.24, 0.26, −0.063, 0.13, and 0.072 respectively. The highest value of the Metal-N bond indicates that it has the maximum and direct correlation with C=O. In contrast, the smaller values for the other bonds mentioned above show their trivial impact on C=O.

Moreover, metal-NCS bond shows 0.18, 0.052, −0.47, −0.063, 0.16, with C-C bond, C=C bond, C-N bond, C=O bond, C-O bond, respectively. All the positive values are too small to impact each other. However, the value −0.99 reported in the sixth row and eighth column of fig 5 indicates the highest inverse correlation of the metal-NCS bond with the Metal-N bond. The presence of one such bond is a strong hindrance for another bond in the same dye.

Next, the C-O bond shows 0.11, −0.065, −0.19, 0.13, 0.16, −0.15 with C-C bond, C=C bond, C-N, C=O, metal-NCS bond, and metal-N bond respectively. This indicates the minimum impact of these bonds on the presence of a C-O bond in a dye.

Further, the Metal-N bond shows −0.14, −0.017, 0.49, 0.072, −0.99, −0.15 with C-C bond, C=C bond, C-N, C=O, metal-NCS bond, C-O bond respectively. It is clear from these values that the Metal-N bond has the maximum direct correlation with the C=O bond, and the highest inverse correlation with the metal-NCS bond.

Based on the above interpretation, it is obvious that the highest direct correction value of $\lambda_{max}$ is 0.77, observed with metal-NCS bonds. Whereas the maximum inverse correlation

is $-0.74$, observed with metal-N bonds. Therefore, if we want to fabricate a dye with a higher value of $\lambda_{max}$, the dye with a greater number of metal-NCS bonds should be fabricated and vice versa.

### 3.2. Prediction of $\lambda_{max}$ Based on $N^+Bu_4$, O-H, O-Na, C-S, and C-Se Bonds

The correlation between $\lambda_{max}$ and other bonds viz. $N^+Bu_4$=2, O-H=1, O-Na=1, O-H=2 C-S=4, C-S=8, and C-Se=4 present in a dye molecule was also studied. The correlation of these bonds with the value of $\lambda_{max}$ as well as a favor for co-existence of these bonds are shown in Figure 6. The highest correlation of $\lambda_{max}$ is 0.19 is observed in the first row and fifth column when two O-H groups are present in a dye. The positive correlation of 0.13, 0.1, 0.15, 0.19, 0.045 is observed between $\lambda_{max}$ and other bonds $N^+Bu_4$=2, O-H=1, O-Na=1, O-H=2 and C-Se=4 respectively. In contrast, a negative correlation of $-0.0068$, $-0.065$ exists with bond C-S=4, C-S=8 respectively. So, it is evident from the results reported in Figure 6 that the above-stated bonds have a negligible impact on the value of $\lambda_{max}$. Therefore, the study of a number of these bonds can be ignored while fabricating a dye with the desired value of $\lambda_{max}$.

More analysis of the results given in Figure 6 shows that the maximum correlation of 0.87 is observed in the number of O-H and O-Na groups. However, these also have a negligible impact on the value of $\lambda_{max}$.

### 3.3. Prediction of $\lambda_{max}$ Based on C-S, $N^+(C_4H_9)$, O-H, C-F, $N{\equiv}N$ Bonds

Now, the correlation of absorption maxima with the bonds such as twelve C-S groups, one $N^+(C_4H_9)$, two C-S, three O-H, three C-F, one $N{\equiv}N$, six C-F was also studied as shown in Figure 7. The analysis of results given in Figure 7 shows that the $\lambda_{max}$ is negatively correlated with C-S, O-H, C-F, $N{\equiv}N$ with a negligible impact. On the other hand, the $\lambda_{max}$ is positively correlated with $N^+(C_4H_9)$ and C-S groups with values of 0.034 and 0.37. These values indicate that the C-S group has the highest impact on the value of $\lambda_{max}$. However, the degree of correlation is not very significant. The remaining groups have a negligible impact on the value of $\lambda_{max}$.

Further, it is clear from the results shown in Figure 7 that C-F, O-H, and $N{\equiv}N$ bonds favor their coexistence in a dye. Therefore, while fabricating a dye with the desired value of $\lambda_{max}$, the number of C-S groups should be considered. Furthermore, the number of C-F, O-H, and $N{\equiv}N$ bonds can be increased or decreased in proportion to each other.

### 3.4. Prediction of $\lambda_{max}$ Based on O-H, $N{\equiv}N$, $TBA^+$, N-H Bond

Now, the correlation of $\lambda_{max}$ with different bonds such as O-H=7, $N{\equiv}N$=2, $TBA^+$=1, N-H=2, N-H=4, O-H=4, and $TBA^+$=1 was also studied. It has been observed that only the presence of one $TBA^+$ group in a dye have a significant and direct correlation with the value of $\lambda_{max}$. Increasing the number of $TBA^+$ groups can result in the dye with a higher value of $\lambda_{max}$. However, there is a negligible direct impact of the N-H group on the value of $\lambda_{max}$.

Similarly, there is an insignificant inverse impact of O-H=7, $N{\equiv}N$=2, O-H=4 groups on the value of $\lambda_{max}$. Furthermore, it is clear from the values reported in Figure 8 that the aforementioned bonds do not favor or hinder their co-existence.

### 3.5. Prediction of $\lambda_{max}$ Based on Atomic and Molecular Weight

The value of $\lambda_{max}$ is also dependent on the atomic mass of an atom present in a dye molecule. Furthermore, it is dependent on the molecular mass of a molecule present in a dye and the complete molecule of a dye. The inter-correlation of $\lambda_{max}$ with the individual atomic masses of Carbon (C), Hydrogen (H), Nitrogen (N), Oxygen (O), Ruthenium (Ru), Sulfur (S), Sodium (Na), Selenium (Se), Fluorine (F) is demonstrated in Figure 9. Furthermore, the directly or inversely correlated atoms or groups of a dye are presented in Figure 9.

The absorption maxima show a negative correlation with an atomic weight of N and F in proportion to values $-0.32$ and $-0.58$ respectively. However, it shows a positive

correlation with C, H, O, S, Na and molecular weight of the dye in proportion to the values 0.16, 0.36, 0.098, 0.44, 0.16, and 0.17 respectively. It is obvious from these values that increasing the mass of Sulfur in a dye leads to a significant increase in the value of $\lambda_{max}$.

Increasing the mass of Fluorine may lead to a decrease in the value of $\lambda_{max}$. Thus, it is apparent that if we want to fabricate the dye with a higher value of $\lambda_{max}$ then the number of Sulfur atoms in a dye molecule must be increased. On the other hand, if we want to fabricate a dye with a lower value of $\lambda_{max}$, then the number of F atoms in a dye molecule must be increased. The other atoms, viz. C, H, N, O, Ru, Na, and Se, have a negligible impact on the value of $\lambda_{max}$. Furthermore, the molecular weight of a dye has an insignificant impact on the value of $\lambda_{max}$. The impact is in proportion to the value of 0.17 only.

Further, it is evident from the results reported in Figure 8 that the atomic weight of C shows a correlation to H, N, O, S, Na, F with values 0.75, 0.37, 0.14, 0.26, 0.26, and 0.34, respectively. The C atom is in a strong correlation of 0.95 with the molecular weight of a dye. This shows that a greater number of C atoms are present in a dye with high molecular weight and vice versa.

Similarly, the atomic weight of the H atom shows a correlation with C, N, O, S, Na, and F in proportion to values 0.75, 0.21, 0.32, 0.23, −0.043, and −0.42 respectively. These values indicate that C and H atoms significantly favor their co-existence in a dye molecule. Whereas H and F atoms hinder the co-existence of each other. Other atoms, viz. N, O, S, and Na, have a negligible impact on the presence of H atoms in a dye molecule. Further, the H atom shows a correlation of 0.78 with the molecular weight of a dye. The higher molecular weight of a dye favors the presence of a greater number of H atoms in it.

Next, it is evident from Figure 8 that the atomic weight of N is correlated to atomic weights of C, H, O, S, Na, and F, with values 0.37, 0.21, 0.094, 0.36, −0.0066, and 0.39, respectively. These values are too small to have any significant effect on each other. Further, it is correlated to the molecular weight of dye in proportion to the value 0.42.

Similarly, the molecular weight of the O atom is also correlated with the molecular weight of C, H, N, S, Na, Se, F in proportion to values 0.14, 0.32, 0.094, −0.021, 0.24, −0.074, and −0.028, respectively. These values indicate that the O atom has minimum interference with the presence of the other atoms in a dye molecule. Moreover, the molecular weight of a dye also has a minimum correlation of 0.098 with the presence of an O atom in a dye molecule.

Further, the atomic weight of the S atom shows values 0.26, 0.23, −0.36, −0.021, 0.073, −0.036, −0.53, for the atomic weight of C, H, N, O, Na, Se, and F, respectively. Its correlation with the molecular weight of a dye is observed as 0.33. These values show that the atomic weight of S is inversely correlated with the atomic weight of N, O, Se, and F atoms but the degree of correlation is not significant. It is, however, directly correlated to atomic weights of C, H, and Na. Furthermore, the degree of direct correlation is insignificant.

The atomic weight of Na also shows a correlation with the atomic weight of C, H, N, O, Ru, Se, S, and F with values 0.16, 0.026, −0.043, −0.0066, 0.24, 0.073, −0.032, −0.15, respectively. It is apparent from these values that the direct, as well as inverse correlation of atomic weight of Na with above-stated atoms, is negligible.

Now, it has been observed from Figure 9 that the atomic weight of Se is correlated with C, H, N, O, S, Na, and F with values −0.066, −0.046, −0.1, −0.074, −0036, −0.032, −0.061, respectively. The low positive, as well as negative values, clearly show that the presence of the Se atom in a dye molecule is not determined by the presence of other atoms.

Similarly, the atomic weight of F is correlated with the atomic weights of C, H, N, O, S, Na, and Se with the values of −0.34, −0.42, 0.39, −0.028, −0.53, −0.15, −0.061, respectively. These values indicate that the presence is F is inversely related to all the above-mentioned atoms except N. However, the degree of correlation is not very high.

Further, the molecular weight of the dye is also correlated with absorption maxima and atomic weights of C, H, N, O, S, Na, Se, and F in proportion to the values 0.95, 0.78,

0.42, 0.27, 0.33, 0.054, −0.023, −0.25, respectively. It indicates that the molecular weight is highly dependent on the atomic weight of carbon and H.

### 3.6. Difference and Percentage Error

To validate the reliability and accuracy of the proposed model, we calculated the difference in the experimental values of $\lambda_{max}$ reported in the literature and the predicted values [30,31]. Furthermore, we calculated the percentage error in the experimental and predicted values of $\lambda_{max}$. The values of difference and percentage error of individual dyes are demonstrated in Table 2. Its first column shows the name of the dye, the second column shows the experimental value of $\lambda_{max}$, the third column includes the predicted value of $\lambda_{max}$, the fourth column shows the difference in $\lambda_{max}$ values, and the last column contains the values of percentage error.

**Table 2.** Comparison of experimental and predicted values of absorption maxima.

| Dye | $\lambda_{max}$ (Experimental) | $\lambda_{max}$ (Predicted) | Difference | Percentage Error | *t*-Score | Ref. |
|---|---|---|---|---|---|---|
| N749 | 600 | 599.9605103 | −0.039489746 | 0.006581625 | 2.962804 | [48] |
| N719 | 525 | 525.0811157 | 0.081115723 | 0.015450614 | 2.573224 | [52] |
| Z907 | 520 | 518.3882446 | −1.611755371 | 0.309952945 | 2.554664 | [51] |
| YS-1 | 536 | 535.9921265 | −0.007873535 | 0.001468943 | 2.633939 | [51] |
| YS-2 | 536 | 536.6464233 | 0.64642334 | 0.120601371 | 2.637666 | [51] |
| YS-3 | 539 | 538.6637573 | −0.336242676 | 0.062382687 | 2.65839 | [51] |
| YS-4 | 535 | 534.9170532 | −0.082946777 | 0.015504071 | 2.642976 | [51] |
| YS-5 | 555 | 554.8757324 | −0.124267578 | 0.022390554 | 2.746662 | [51] |
| CYC-B1 | 553 | 554.9368896 | 1.936889648 | 0.350251287 | 2.737349 | [53] |
| CYC-B3 | 544 | 543.5513306 | −0.448669434 | 0.082475998 | 2.699216 | [54] |
| SJW-E1 | 546 | 545.9083252 | −0.091674805 | 0.016790258 | 2.713337 | [54] |
| C101 | 547 | 545.3406372 | −1.659362793 | 0.303357005 | 2.72482 | [55] |
| C102 | 547 | 545.6287842 | −1.37121582 | 0.250679314 | 2.728966 | [55] |
| C103 | 550 | 549.0956421 | −0.90435791 | 0.164428711 | 2.74789 | [56] |
| C104 | 553 | 554.3543701 | 1.354370117 | 0.24491322 | 2.764281 | [57] |
| C105 | 550 | 546.0117188 | −3.98828125 | 0.725142062 | 2.759501 | [58] |
| C106 | 550 | 549.9251099 | −0.074890137 | 0.013616389 | 2.759075 | [55] |
| C107 | 559 | 558.5645142 | −0.43548584 | 0.07790444 | 2.808949 | [56] |
| K19 | 545 | 545.203064 | 0.203063965 | 0.037259445 | 2.740354 | [59] |
| K77 | 546 | 544.6682739 | −1.331726074 | 0.243905872 | 2.752063 | [60] |
| CYC-B11 | 554 | 552.9317627 | −1.068237305 | 0.19282262 | 2.796931 | [61] |
| CYC-B6L | 551 | 545.9569702 | −5.043029785 | 0.915250421 | 2.790254 | [62] |
| CYC-B6S | 548 | 547.302124 | −0.697875977 | 0.12734963 | 2.774375 | [62] |
| CYC-B7 | 551 | 552.8936157 | 1.893615723 | 0.343668908 | 2.791177 | [63] |
| CYC-B13 | 547 | 548.375 | 1.375 | 0.251371115 | 2.775524 | [64] |
| JK-55 | 539 | 538.1456909 | −0.854309082 | 0.158498898 | 2.74214 | [65] |
| JK-56 | 537 | 538.4175415 | 1.417541504 | 0.26397422 | 2.734822 | [65] |
| RC-31 | 560 | 558.7681274 | −1.231872559 | 0.219977245 | 2.860881 | [66] |
| RC-32 | 564 | 562.3169556 | −1.683044434 | 0.298412144 | 2.884413 | [66] |
| RC-36 | 547 | 545.4786377 | −1.521362305 | 0.278128386 | 2.798735 | [66] |
| PRT1 | 520 | 520.6152954 | 0.61529541 | 0.118326038 | 2.663049 | [67] |
| PRT2 | 517 | 517.5576172 | 0.557617188 | 0.107856326 | 2.652854 | [67] |
| PRT3 | 519 | 519.3753052 | 0.375305176 | 0.072313137 | 2.668113 | [67] |
| PRT4 | 519 | 514.8287964 | −4.171203613 | 0.803700089 | 2.678184 | [67] |
| PRT21 | 514 | 515.2487793 | 1.248779297 | 0.242953166 | 2.651222 | [68] |
| PRT22 | 520 | 519.6818237 | −0.31817627 | 0.061187744 | 2.688368 | [68] |
| PRT23 | 536 | 537.9351196 | 1.935119629 | 0.361029774 | 2.774103 | [68] |
| PRT24 | 537 | 537.1176147 | 0.117614746 | 0.021902187 | 2.787605 | [68] |
| TF1 | 510 | 509.9632568 | −0.036743164 | 0.007204542 | 2.653727 | [69] |
| TF2 | 509 | 510.1255798 | 1.125579834 | 0.221135527 | 2.651082 | [69] |
| TF3 | 513 | 513.4284058 | 0.428405762 | 0.083509892 | 2.676337 | [69] |
| TF4 | 516 | 516.3215942 | 0.321594238 | 0.062324464 | 2.69655 | [69] |

**Table 2.** *Cont.*

| Dye | $\lambda_{max}$ (Experimental) | $\lambda_{max}$ (Predicted) | Difference | Percentage Error | *t*-Score | Ref. |
|---|---|---|---|---|---|---|
| MJ-4 | 594 | 594.5967407 | 0.596740723 | 0.100461401 | 3.109695 | [70] |
| MJ-6 | 608 | 606.5596313 | −1.440368652 | 0.236902744 | 3.174494 | [70] |
| MJ-7 | 603 | 606.5596313 | 3.559631348 | 0.590320289 | 3.119411 | [70] |
| MJ-10 | 630 | 630.5605469 | 0.560546875 | 0.088975698 | 3.233595 | [70] |
| MJ-11 | 630 | 630.5605469 | 0.560546875 | 0.088975698 | 3.149689 | [70] |
| MJ-12 | 631 | 632.057312 | 1.057312012 | 0.167561337 | 3.010226 | [70] |
| TFRS-1 | 532 | 534.241333 | 2.241333008 | 0.421303183 | 2.251667 | [71] |
| TFRS-2 | 533 | 533.7162476 | 0.716247559 | 0.1343804 | 2.269726 | [71] |
| TFRS-3 | 503 | 503.7233887 | 0.723388672 | 0.143814847 | 2.151846 | [71] |
| TFRS-4 | 501 | 501.8835449 | 0.883544922 | 0.176356271 | 2.146519 | [71] |
| TFRS-21 | 499 | 500.2005005 | 1.200500488 | 0.240581259 | 2.137079 | [72] |
| TFRS-22 | 473 | 473.5461426 | 0.546142578 | 0.115463547 | 2.022101 | [72] |
| TFRS-24 | 485 | 485.4335938 | 0.43359375 | 0.089400776 | 1.963301 | [72] |
| TFRS-51 | 499 | 499.3976135 | 0.397613525 | 0.079682067 | 1.925546 | [73] |
| TFRS-52 | 495 | 495.3208618 | 0.320861816 | 0.064820565 | 1.866991 | [73] |
| TFRS-53 | 500 | 500.4888611 | 0.488861084 | 0.097772218 | 1.795062 | [73] |
| TFRS-54 | 496 | 496.4121094 | 0.412109375 | 0.083086565 | 1.695218 | [73] |
| CS9 | 518 | 518.0759277 | 0.075927734 | 0.014657863 | 1.573186 | [74] |
| A597 | 539 | 538.2700806 | −0.729919434 | 0.135421053 | 1.632301 | [75] |
| CS27 | 517 | 517.1707764 | 0.170776367 | 0.033032179 | 1.590288 | [76] |
| CS28 | 518 | 517.486084 | −0.513916016 | 0.099211589 | 1.579045 | [76] |
| CS32 | 518 | 518.0393066 | 0.039306641 | 0.007588155 | 1.557655 | [76] |
| CS43 | 518 | 518.1796875 | 0.1796875 | 0.034688707 | 1.529054 | [76] |
| CS17 | 530 | 529.7614136 | −0.238586426 | 0.045016307 | 1.521574 | [76] |
| CS22 | 533 | 532.6157837 | −0.384216309 | 0.072085612 | 1.578218 | [76] |
| LXJ-1 | 549 | 548.1496582 | −0.850341797 | 0.154889211 | 1.686294 | [56] |
| KW-1# | 515 | 514.050293 | −0.949707031 | 0.184409127 | 1.566551 | [77] |
| KW-2# | 550 | 549.9004517 | −0.09954834 | 0.018099697 | 1.529517 | [77] |
| HRD-1 | 543 | 542.3900757 | −0.609924316 | 0.112324923 | 1.449902 | [78] |
| K-73 | 545 | 545.2966309 | 0.296630859 | 0.05442768 | 1.48967 | [79] |
| KC-5# | 537 | 537.1211548 | 0.121154785 | 0.022561412 | 1.454641 | [80] |
| KC-6# | 531 | 527.5883179 | −3.411682129 | 0.642501354 | 1.539624 | [80] |
| KC-7# | 533 | 530.7481079 | −2.25189209 | 0.422493815 | 1.649544 | [80] |
| KC-8 | 522 | 522.1340332 | 0.134033203 | 0.02567686 | 1.730241 | [80] |
| MH06 | 541 | 540.6497192 | −0.350280762 | 0.064746909 | 1.841917 | [81] |
| MH11 | 547 | 546.6559448 | −0.344055176 | 0.062898569 | 1.987435 | [81] |
| MC119 | 548 | 548.0170898 | 0.017089844 | 0.003118585 | 1.976855 | [82] |
| S3 | 516 | 515.6741333 | −0.325866699 | 0.063152462 | 1.256043 | [83] |
| S4 | 518 | 519.6868286 | 1.686828613 | 0.325642586 | 1.462133 | [83] |

# These compounds were named after the initials of the first author of the reference cited.

It is evident from the results shown in Table 2 that the predicted values of $\lambda_{max}$ are closer to the experimental values of $\lambda_{max}$ collected from the literature. The range of difference in predicted and experimental values is −5.04 to 3.559, which is very low. Furthermore, the percentage error observed in the experimental and predicted values lie is in the range of 0.00145 to 0.915. The low values prove the reliability of the model.

Statistical Analysis: For statistical analysis of the results shown in Table 2, we applied the *t*-test as defined in Equation (4). The value of *t* as shown in column 6 of Table 2 lies in the range of 1.25 to 3.25. A positive absolute value of *t* indicates a moderate difference between the means of the two sets of values. It also suggests that the difference between the means is statistically significant, implying that the predicted values significantly deviate from the experimental values.

$$t - score = \frac{Difference\ between\ experimental\ and\ predicted\ value}{standard\ error} \qquad (4)$$

## 4. Discussion

The research work proposed in this work meets the objectives of accurately predicting the absorption maxima. The ANN model designed in this research correctly predicts the value of absorption maxima of the dye sensitizer based on the structure of the dye molecule, numbers of bonds, molecular weight, and atomic weight. The model, trained with different sets of values of these parameters and their corresponding value of the compound, learns to predict the correct value of the absorption maxima. It magnificently reduced the difference and percentage error in the experimental and predicted values of $\lambda_{max}$ of a dye, reported in the literature [30,31]. Furthermore, it predicted the impact of collective atomic weights of each atom type and molecular weight of the dye molecule on the value of $\lambda_{max}$ of a dye. Moreover, the proposed research work successfully showed the impact of different bonds between constituent atoms of a dye on its absorption maxima.

To achieve the first objective ANN model is trained to predict the absorption maxima of inorganic dye mainly ruthenium complexes used in DSSC. Further, the calculated values of absorption maxima are very close to the experimental values collected from the literature as shown in Table S1. Moreover, it reported the lesser difference in experimental and calculated values of absorption maxima reported by Xu et.al. [30,31]. They reported the difference in range of $-27.3$ to $27.1$ [30] and $-16.6$ to $16.2$ [31]. In contrast, the difference reported in the work proposed in this manuscript lies in the range of $-5.043029785$ to $3.559631348$. This shows that the proposed model more accurately predicts the value of $\lambda_{max}$. Moreover, the statistical analysis of the difference in the experimental and predicted values in terms of the *t*-test validates the obtained result. The *t*-score obtained in the range of $1.25$ to $3.23$ signifies the importance of the *t*-test.

In addition, the percentage error observed from the experimental results lies in the range of $0.01468943$ to $0.915250421$, which are $96.63$ and $94.35\%$ less than the percentage error calculated in the works reported in [30,31].

To demonstrate the impacts of the atomic weight of each atom and molecular weight on the value of $\lambda_{max}$, a correlation matrix was studied as shown in Figure 9. The atomic weight of sulfur and fluorine shows a considerable impact on $\lambda_{max}$ value. The $\lambda_{max}$ value is directly proportional to the sulfur atoms, whereas it is indirectly proportional to the fluorine atoms.

Further, the correlation matrix of $\lambda_{max}$ with different bonds was studied as shown in Figures 5–8. Initially, the correlation matrix of $\lambda_{max}$ with C-C, C=C, C-N, C=O, Metal-NCS, C-O, Metal-N bonds as demonstrated in Figure 5 shows that the $\lambda_{max}$ directly depends only on the Metal-NCS bond and inversely depends on Metal-N bonds. The next correlation matrix as shown in Figure 6 indicates the relation of $\lambda_{max}$ and N+Bu4=2, O-H=1, O-Na=1, O-H=2 C-S=4, C-S=8, and C-Se=4. It is concluded that the $\lambda_{max}$ directly depends on the bond O-H=2, whereas it inversely depends on the C-S=8 bond. Likewise, the correlation of $\lambda_{max}$ based on C-S, N$^+$(C4H9), O-H, C-F, N≡N bonds were studied. It is observed that $\lambda_{max}$ directly depends on the C-S bonds and is inversely dependent on the C-F bond. Furthermore, the correlation of $\lambda_{max}$ based on O-H, N-N, TBA$^+$, N-H bonds were studied. It is concluded that the impact on the $\lambda_{max}$ is observed at its maximum with the increase in the TBA+ group in dye structure, whereas the $\lambda_{max}$ value decreases with an increase in N≡N bonds.

The impacts of activation function 'ReLU', loss function Mean Absolute Error (MSE), and optimizer 'Adam' on the prediction accuracy of $\lambda_{max}$ using the ANN model were observed. It is concluded that the proposed model is more efficient than the methods available in the literature [30,31]. It accounts for the diminishing difference in predicted and experimental values of $\lambda_{max}$. Furthermore, justify the relation of $\lambda_{max}$ with other bonds of dye molecule as well as the impact of atomic and molecular weight on it.

Further, the superiority of the customized ANN model over the traditional machine learning models is proved by conducting the experiments. The values of Mean Square Error (MSE), and Percentage Error reported by ANN, Xgboost, and Random Forest models were recorded for each epoch. These values are illustrated in Figures 10 and 11, respectively.

This is evident from Figure 10, that the ANN model reports the lowest value of MSE among all the three models. This proves that ANN model outperforms the Xgboost and Random Forest in predicting the value of absorption maxima.

Similarly, the results demonstrated in Figure 11 prove that the ANN model reports the lowest value of percentage error among all the above-stated three models. This justifies the efficacy of ANN model in correctly predicting the value of absorption maxima.

## 5. Conclusions

In this manuscript, a customized ANN model is developed for automating the prediction of the value of absorption maxima of dye without actually fabricating the dye. Further, the ANN model is fine-tuned to minimize the difference in the experimental and predicted values. The model precisely predicted values of absorption maxima. The difference reported in the proposed work is $-22.3$ to 23.6 values lower than the range of difference reported in [30,31]. Similarly, the difference is lower by values from 11.56 to 12.7. This proves the supremacy of the proposed work over the reported methods in the literature.

Furthermore, the research work available so far does not focus on showing the impact of the atomic weight of atoms in a dye molecule, the number, and types of bonds available in a dye molecule. The research works presented in this manuscript showcase the role of atomic weights and different types of bonds present in dye molecules on the value of their absorption maxima. Furthermore, they show the direct and inverse correlation of individual atoms present in a dye molecule on the value of absorption maxima. Moreover, they predict the inter-correlation among different atoms present in a dye molecule. The experimental results prove the efficacy of the proposed work, minimizing the requirements for hit and trial experiments. Therefore, it is a cost-saving approach for fabricating the dye of desired characteristics.

The model is purely dependent on the descriptors originated from the chemical structure of the dye molecule and valid for regular dyes of whichever chemical structure. Therefore, this model would be beneficial for the synthesis of new sensitizers with preferred absorption maxima values for DSSC.

**Supplementary Materials:** The following are available online at https://www.mdpi.com/article/10.3390/bdcc7020115/s1, Table S1: Dataset.

**Author Contributions:** All authors of this manuscript have equal contribution in conceptualization, methodology, software, validation, formal analysis, investigation, data curation, writing—original draft preparation, writing—review and editing, and visualization. N.T. and K.G. are involved in implementation and conducting experiments, G.R., V.S.D. and P.K.S. are also involved in supervision. E.Z. and E.V. are also involved in project administration, and funding acquisition. All authors have read and agreed to the published version of the manuscript.

**Funding:** This research is funded by DIMES (Department of Informatics, Modeling, Electronics and Systems)—University of Calabria with Grant/Award Number: SIMPATICO_ZUMPANO.

**Data Availability Statement:** The sample dataset used in this research is submitted as Supplementary Material. The code is available with authors. They will provide the code on request.

**Acknowledgments:** The authors would like to acknowledge the financial support by the Science and Engineering Research Board (SERB) under Grant No. [EMR/2016/006259].

**Conflicts of Interest:** The authors declare no conflict of interest.

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
