# Peer review of "Molecular Structure-Based Prediction of Absorption Maxima of Dyes Using ANN Model"

_2504-2289, doi:10.3390/bdcc7020115_

Round 1

Reviewer 1 Report

This paper proposed an ANN-based QSPR model for correctly predicting the value of λmax for inorganic ruthenium complex dyes used in DSSC.

However, there are comments for improving the paper's quality.

1. This work is still missing for comparison with other approaches. To handle this problem, many traditional machine-learning models can predict continuous variables, such as random forest and xgboost. Please consider this comment to improve paper quality.

2. On table 1, the paper should show the statistical report rather than the output of the model.

1. Please improve the figure quality. Some of them have low-quality figures.

2. Please explain in more detail the caption of figures and tables.

Author Response

. This work is still missing for comparison with other approaches. To handle this problem, many traditional machine-learning models can predict continuous variables, such as random forest and xgboost. Please consider this comment to improve paper quality.

Response: We implemented the machine learning algorithms “XGBoost” and Random forest. The results have been added to the discussion section of the manuscript. The result analysis is added as Figure 10 and Figure 11 in the manuscript.

  1. On table 1, the paper should show the statistical report rather than the output of the model.

Response: Now, the table number is updated to 3. In this table the value of t-score is also added. The interpretation of the t value is added to the manuscript below table 3.  

  1. Please improve the figure quality. Some of them have low-quality figures. Please explain in more detail the caption of figures and tables.

Response: As per the comment, the figure quality is improved. Figures 5 to 9 have been updated with improved quality.

Reviewer 2 Report

The article entitled “Molecular structure-based prediction of absorption maxima of dyes using ANN model” is well-written and, from my point of view, would be of interest for the readers of Big Data and Cognitive Computing. In spite of this, and before its publication, I would like to suggest authors to perform the following changes:

Introduction: please include a short paragraph in which the structure of the article is described.

Lines 143-144: it is said ‘To prepare the dataset, the molecular structures of 81 ruthenium dye complexes were taken from literature’. From my point of view, references should be included in this sentence.

Lines 152-153 it is said ‘This large variation in the range of values is important for improving the robustness of the ANN model’, please explain the reason of this fact.

Section 2.2.1. Please give some details about the artificial neural network model. I suposse this is a back-propagation neural network.

Line 173: it speaks about activation funcionts. What is an activation function? What is the equation of, at least, the one selected? The same can be applied to the errors.

In the Figure 5, correlation matrix, numbers are not clear enough. Please reduce font size. The same can be applied to Figures form 6 to 9.

In lines 426-427 it is said ‘The research work proposed in this work meets the objectives of accurately predicting the absorption maxima’. From my point of view, after such sentence, an explanation of the reasons for such assessment is required.

Author Response

Reviewer-2

The article entitled “Molecular structure-based prediction of absorption maxima of dyes using ANN model” is well-written and, from my point of view, would be of interest for the readers of Big Data and Cognitive Computing. In spite of this, and before its publication, I would like to suggest authors to perform the following changes:

Introduction: please include a short paragraph in which the structure of the article is described.

Response: In response to the comment, the paragraph stating the structure of the article is added to the introduction section of the manuscript. For your ready reference, it is also given below.

The structure of the article is as follows: Section 1 provides the introduction. It gives an overview of the research topic, introduces the research problem, highlights the gaps in existing knowledge, and presents the objectives of the study. Section 2 describes the data collection and methodology of the research work. Section 3 illustrates the results. It presents the findings of the study. It includes tables, figures, and statistical analyses to support the findings. Section 4 presents the discussion of the research work. It interprets and analyzes the results, relates them to the objectives, and compares them with previous research. Section 5 presents the conclusion of the work. It summarizes the main findings and their implications. It also offers insights for future investigations.

Lines 143-144: it is said ‘To prepare the dataset, the molecular structures of 81 ruthenium dye complexes were taken from literature’. From my point of view, references should be included in this sentence.

Response: The dataset collected is uploaded as supplementary material in table 2. The reference for each data item is given in table 2.

Lines 152-153 it is said ‘This large variation in the range of values is important for improving the robustness of the ANN model’, please explain the reason of this fact.

Response: In response to the comment, we have added the reason for robustness in lines 162 to 165 of the manuscript. It is also written below for your ready reference.

This large variation in the range of values is important for improving the robustness of the ANN model. It means that the performance of the model does not degrade with change in the value of λmax.  So, the model work efficiently for a wide range of dyes to correctly predict the value of λmax.

Section 2.2.1. Please give some details about the artificial neural network model. I suppose this is a back-propagation neural network.

Response: In response to the comment, the details of the ANN model have been added to the manuscript from line 175 to 183. The details have also been added below for your ready reference.

An Artificial Neural Network (ANN) is a machine learning model that is inspired by the structure and function of biological neurons in the brain. An ANN consists of multiple interconnected nodes i.e. neurons, organized into layers. Each neuron in the network has a set of weights associated with it, which determine the strength of its connections to other neurons in the network. The input layer of an ANN receives input data, which is then passed through one or more hidden layers. Each hidden layer applies a non-linear transformation to the input. During the training phase, its neurons adjust the weights to minimize the difference (value of loss function) between the predicted output and the actual output. Finally, the output layer of the network provides the prediction.

Line 173: it speaks about activation function. What is an activation function? What is the equation of, at least, the one selected? The same can be applied to the errors.

Response: The details of the activation function have been added from line 230 to 238 in the manuscript. The details are also given here for your reference.

Activation functions are employed in the neural networks to introduce non-linearity and enabling them to learn complex patterns in the input data. In this research, we employed the ReLU (Rectified Linear Unit) activation function. It is a simple and computationally efficient function that sets all negative values in the input to zero and leaves positive values unchanged as defined in equation given below.

Here, x is the input to the function, and f(x) is the output. The ReLU function returns the input x, if it is positive, and returns 0 otherwise. This makes the ReLU function a simple yet powerful way to introduce non-linearity into neural networks.

In Figure 5, correlation matrix, numbers are not clear enough. Please reduce font size. The same can be applied to Figures from 6 to 9.

Response: Figures 5 to 9 have been updated for clear reading. the quality of figures have been improved.

In lines 426-427 it is said ‘The research work proposed in this work meets the objectives of accurately predicting the absorption maxima’. From my point of view, after such sentence, an explanation of the reasons for such assessment is required.

Response: To develop an ANN-based model for predicting the absorption maxima of the dye sensitizer used in DSSC. The reason for the statement is added to the manuscript in line 470 to 475. The reason is also given below for your reference.

The ANN model designed in this research correctly predicts the value of absorption maxima of the dye sensitizer based on the structure of the dye molecule, number of bonds, molecular weight, and atomic weight of the dye molecule. The model trained with different sets of values of the above-stated parameters and their corresponding value of compound learns to predict the correct value of the absorption maxima.

Round 2

Reviewer 1 Report

Thank you for the response.

All comments have been addressed in the current version.

Thank you.